# Identification of Trehalose-6-Phosphate Synthase (TPS) Genes Associated with Both Source-/Sink-Related Yield Traits and Drought Response in Rapeseed (*Brassica napus* L.)

**DOI:** 10.3390/plants12050981

**Published:** 2023-02-21

**Authors:** Bo Yang, Liyuan Zhang, Sirou Xiang, Huan Chen, Cunmin Qu, Kun Lu, Jiana Li

**Affiliations:** 1Chongqing Rapeseed Engineering Research Center, College of Agronomy and Biotechnology, Southwest University, Chongqing 400716, China; 2Academy of Agricultural Sciences, Southwest University, Chongqing 400716, China

**Keywords:** trehalose-6-phosphate synthase, phylogenetic analysis, expression pattern, yield, drought resistance, *Brassica napus*

## Abstract

Trehalose-6-phosphate synthase (TPS) is an important enzyme for the synthesis of Trehalose-6-phosphate (T6P). In addition to being a signaling regulator of carbon allocation that improves crop yields, T6P also plays essential roles in desiccation tolerance. However, comprehensive studies, such as evolutionary analysis, expression analysis, and functional classification of the *TPS* family in rapeseed (*Brassica napus* L.) are lacking. Here, we identified 35 *BnTPS*s, 14 *BoTPS*s, and 17 *BrTPS*s in cruciferous plants, which were classified into three subfamilies. Phylogenetic and syntenic analysis of *TPS* genes in four cruciferous species indicated that only gene elimination occurred during their evolution. Combined phylogenetic, protein property, and expression analysis of the 35 *BnTPS*s suggested that changes in gene structures might have led to changes in their expression profiles and further functional differentiation during their evolution. In addition, we analyzed one set of transcriptome data from Zhongshuang11 (ZS11) and two sets of data from extreme materials associated with source-/sink-related yield traits and the drought response. The expression levels of four *BnTPS*s (*BnTPS6*, *BnTPS8*, *BnTPS9*, and *BnTPS11*) increased sharply after drought stress, and three differentially expressed genes (*BnTPS1*, *BnTPS5*, and *BnTPS9*) exhibited variable expression patterns among source and sink tissues between yield-related materials. Our findings provide a reference for fundamental studies of *TPS*s in rapeseed and a framework for future functional research of the roles of BnTPSs in both yield and drought resistance.

## 1. Introduction

Trehalose is a nonreducing disaccharide [1] that widely exists in distinct organisms, such as fungi, algae, bacteria, and plants [2,3,4]. Trehalose is important for the growth and development of plants and for protecting plants from abiotic stress, i.e., desiccation, heat, freezing, cold, and oxidative stress [3,5,6,7,8]. Resurrection plants accumulate abundant trehalose under desiccation stress to maintain a metabolically static state [9]. Trehalose-6-phosphate synthase (TPS) plays a key role in the synthesis and metabolism of trehalose. The biosynthesis of trehalose in plants involves two enzymatic steps. First, TPS catalyzes the synthesis of trehalose-6-phosphate (T6P) from uridine diphosphate-glucose (UDPG) and glucose 6-phosphate (G6P) [10]. Subsequently, T6P is converted to trehalose by trehalose-6-phosphate phosphatase (TPP), and one molecule of trehalose can be decomposed into two molecules of glucose by trehalase (TRE). Therefore, TPS is a biosynthetic enzyme that catalyzes the formation of T6P, and TPP is a protease that catalyzes the biodegradation of T6P.

TPSs form a large family with multiple copies that show extensive functional diversification [4,11,12,13,14]. The Arabidopsis (*Arabidopsis thaliana*) genome contains 11 *TPS*s and 1 *TRE* [13,15]. The 11 *AtTPS*s are classified into two subfamilies: class A (*AtTPS1–4*) and class B (*AtTPS5–11*). Many *TPS* genes have been identified in plants, including 11 in rice (*Oryza sativa*) [15,16], 12 in *Populus* [14,15], 13 in apple (*Malus domestica*) [17], 53 in cotton (*Gossypium hirsutum*) [18], 8 in potato (*Solanum tuberosum*) [1], and 12 in wheat [19]. Although the gene sequences and deduced amino acid sequences of different *TPS*s are basically the same, numerous studies have revealed functional divergence among different TPS family members. In particular, *TPS* genes from class A and class B have different copy numbers, expression profiles, and functions. Functional complementation of the rice mutants *tps1* and *tps2* showed that only *OsTPS1* encodes an enzyme with TPS activity [20]. Similarly, only *AtTPS1* encodes an enzyme with TPS activity in Arabidopsis [13]. Nevertheless, AtTPS6 regulates plant architecture and cell shape [4]. Some TPSs also act as essential modulators of plant development and inflorescence branching [21]. *OTSA* (*TPS A*) in tobacco influences leaf morphology, growth, and photosynthetic activity [22].

TPS enzymes also function in signaling under biotic and abiotic stress [5,10,23,24]. For example, TPS2 plays important roles as a glucose regulator and also functions in stress signaling in Arabidopsis [25], and *AtTPS1* shows clear responses to abiotic stress as well [10,26,27]. In *Selaginella*, functional studies suggested that SlTPS1 regulates plant responses to salt and heat stress [18,28]. Similarly, overexpressing *TPS* genes in tobacco resulted in pronounced changes in plant growth and morphology under drought stress [23,29].

Rapeseed, one of the most important oilseed crops worldwide, is used to produce oil, animal feed, and healthcare products [30,31]. To date, *TPS*s have been reported in various species, including Arabidopsis, rice, Populus, apple, cotton, and potato [1,12,13,15,17]. However, no genome-wide identification or functional prediction of *TPS*s has been performed in *B. napus*.

In the current study, to examine the evolutionary relationships of *TPS*s in various Brassica species, the genome-wide identification of *TPS*s was performed in four Cruciferae species: *B. napus*, *B. oleracea*, *B. rapa*, and *A. thaliana*. Phylogenetic and syntenic analyses indicated that only gene elimination occurred at the gene level during the genomic evolution of *TPS*s. Analysis of transcriptome data for the 35 *BnTPS*s from yield- and drought-related materials indicated that *TPS*s have undergone considerable functional differentiation during evolution. Among these genes, *BnTPS1*, *BnTPS5*, and *BnTPS9* might be associated with source-/sink-related yield traits and play essential roles in plant growth, whereas other *BnTPS*s (*BnTPS6*, *BnTPS8*, *BnTPS9*, and *BnTPS11*) function in the plant response to drought stress. Additionally, it was revealed by our systematic analysis that *BnTPS9* might be associated with both source-/sink-related yield traits and drought response.

## 2. Results

### 2.1. Identification and Phylogenetic Analysis of the Key TPSs

Based on the deduced amino acid sequences of *TPS*s in Arabidopsis, 35 *BnTPS*s, 14 *BoTPS*s, and 17 *BrTPS*s were identified by both hidden Markov model (HMM) and BLASTp analysis (Table 1). Detailed information about the 35 *BnTPS*s is shown in Appendix A. The isoelectric points (pIs) of the BnTPSs range from 4.81 (*BnTPS7-1*) to 9.49 (*BnTPS5-2*), and their molecular weights (Mw) range from 14.73 (*BnTPS5-4*) to 107.08 (*BnTPS1-2*) (Appendix A).

The 78 *TPS* genes belonging to the four Cruciferae species were divided into three subfamilies based on phylogenetic analysis (Figure 1). Subfamily B contains the most genes, which is consistent with previous findings in Arabidopsis [13].

### 2.2. Syntenic Analysis of TPSs in the Four Cruciferous Species

Syntenic analysis was performed to clarify the evolutionary relationships of the *TPS*s in the four cruciferous species. The syntenic relationships of all the *TPS*s analyzed are shown in Figure 2. In total, 78 homologous pairs were identified (Figure 2A). In addition, collinearity analysis of the *TPS*s from the A and C sub-genomes of *B. napus*, *B. rapa*, and *B. oleracea* and their homologs in Arabidopsis was performed, and 46 and 41 homologous pairs in sub-genomes A (Figure 2B) and C (Figure 2C) were identified, respectively.

The copy numbers of the *TPS* genes between *A. thaliana* and *B. napus* were compared for further evolution analysis (Table 2). Most *BnTPS*s, such as *BnTPS2*, *BnTPS3*, *BnTPS4*, *BnTPS6*, *BnTPS8*, *BnTPS9*, and *BnTPS11*, are present in two copies. All other *BnTPS*s are present in four copies, except for *BnTRE*, which is present in five copies in the genome.

### 2.3. Chromosomal Localization, Gene Structures, and Protein Profiles of BnTPSs

All 33 *BnTPS*s, including 17 in subgenome A and 16 in subgenome C, are unevenly distributed on the 16 chromosomes of *B. napus*, except for chromosomes A04, A05, and C05 (Figure 3). The remaining *BnTPS*s (*BnaAnng13570D*, *BnTPS5-3*; *BnaCnng41250D*, *BnTPS10-3*) are distributed on pseudo-chromosomes Ann and Cnn. Chromosomes A09, A10, C03, and C09 each contain only one *BnTPS*, whereas chromosomes A01, A03, C05, and C07 contain more than three *BnTPS*s (Figure 3).

The exon–intron structures and conserved motifs of the 35 *Bn*TPSs were analyzed based on their protein sequences to obtain additional information about their protein profiles. The number of exons ranged from two (*BnTPS5-2*) to eighteen (*BnTPS1-2*) (Figure 4). In addition, 25 conserved motifs were predicted, most of which were related to the transfer of glycosyl (Figure 5).

These results, combined with the results of phylogenetic analysis, indicate that different genes encode proteins with diverse profiles depending on their phylogenetic relationships. For example, subfamily B genes contain fewer than six exons, whereas subfamily C genes contain eight to eleven exons, and subfamily A genes possess more than 15 exons, indicating that the structures of genes in subfamily A are more complex than the others. Similarly, the proteins from subfamily C contain only two to five motifs, whereas more than 17 motifs were predicted in most proteins from subfamily A and B members besides *BnTPS5-3*/*5-4* and *BnTPS7-1*/*7-2*, pointing to the functional differentiation of subfamily C. Moreover, each copy of the same gene shares similar profiles even though they might be different from those of the other genes in the same subfamily, such as *BnTPS6* vs. *BnTPS10*, and *BnTPS8* vs. *BnTPS9*.

### 2.4. RNA-Seq Analysis and qRT-PCR Verification of the Major Candidate BnTPSs

RNA-Seq data from the 35 *BnTPS*s in *B. napus* cultivar Zhongshuang11 (ZS11) was analyzed to explore their expression specificity, as gene expression patterns are likely associated with gene functions (Appendix A). In general, for most genes in subfamily A, their expression was barely detectable in any tissue across the whole study. Genes in subfamily B exhibited complex expression patterns, whereas *BnTRE*s (subfamily C) tended to be expressed in seeds or pericarps (except for *BnTRE1-1*), and none of these genes showed obvious expression in roots, stems, leaves, or flowers (Figure 6). Moreover, *BnTPS5-1*/*5-2* and *BnTPS7-3*/*7-4* were highly expressed in many tissues, whereas the expression levels of their homologous genes, *BnTPS5-3*/*5-4* and *BnTPS7-1*/*7-2*, were extremely low. Additionally, *BnTPS9-1* and *BnTPS11-1* were expressed at significantly higher levels than the other genes in mature leaves at the full-bloom stage (LeO_f). *BnTPS9-1* was also highly expressed in floral organs, and so was *BnTPS7-3*. As the seeds matured, *BnTPS11-1*/*-2* began to be expressed in seeds collected 40 days after flowering (DAF), and their expression peaked at 49 DAF. On the contrary, the expression of *BnTRE1-2* decreased in the seed coat from 19 to 40 DAF and in seeds from 13 to 49 DAF (Figure 6). These results indicate that *BnTPS* genes have undergone strong functional divergence during their evolution.

Numerous studies have demonstrated that many *TPS* genes are associated with source- and sink-related yield traits and drought response. Therefore, to identify major candidate differentially expressed genes (DEGs) among the 35 *BnTPS*s, two sets of RNA-Seq data in materials with extremely high and low source-/sink-related yields, as well as different levels of drought tolerance were analyzed (Appendix A).

Heat map analysis revealed significantly different expression profiles of the 35 *BnTPS*s among materials with diverse characteristics (Figure 7A,B). For example, genes from subfamily C (*BnTRE*s) showed significantly different expression patterns in the yield-related extreme materials, except for *BnTRE1-1*, whereas all *BnTRE*s were barely expressed in materials that are resistant or sensitive to drought stress (Figure 7A,B). Similarly, *BnTPS6-2* exhibited a lower expression level in yield-related materials, but it was obviously induced after polyethylene glycol (PEG)-6000 treatment in drought resistant (RT) materials, which suggested that *BnTPS6-2* could respond to drought treatment but were not involved in the yield. For almost all samples, the expression level of *BnTPS1-3* was lower in yield-related materials compared with drought-related ones while the other three *TPS1* members were barely expressed in any sample (Figure 7A,B).

Moreover, the expression patterns of different *BnTPS* genes showed spatiotemporal expression specificity even in the same material. In drought-related materials, most genes from subfamily B were expressed at markedly higher levels than those from the two other subfamilies, especially *BnTPS6-1*, *BnTPS6-2*, *BnTPS8-1*, *BnTPS8-2*, *BnTPS9-1*, *BnTPS9-2*, *BnTPS10-1*, *BnTPS10-2*, *BnTPS11-1*, and *BnTPS11-2* (Figure 7B). Likewise, most genes from subfamily A were barely expressed in yield-related materials, whereas *BnTREs* from subfamily C showed higher expression levels, except for *BnTRE1-1* (Figure 7A).

By contrast, the 35 *BnTPSs* also showed significant differential expression depending on their phylogenetic classification even in the same material. For example, *BnTPS5-1* and *BnTPS5-2* were expressed at much higher levels than the other two homologous genes (*BnTPS5-3* and *BnTPS5-4*). Similarly, compared with *BnTPS1-1* and *BnTPS1-4*, *BnTPS1-2* and *BnTPS1-3* showed particularly high expression in sink tissues (seeds at 15 and 35 DAF) (Figure 7A). This phenomenon was more pronounced in subfamily C, where *BnTRE1-1* showed a lower expression level in almost every tissue, but its homologous genes, *BnTRE1-2*, *BnTRE1-3*, *BnTRE1-4*, and *BnTRE1-5*, were expressed at higher levels in specific tissues (Figure 7A). After PEG-6000 treatment, homologous genes of both *BnTPS8* (*BnTPS8-1*/*-2*) and *BnTPS11* (*BnTPS11-1*/*-2*) were induced in the drought-resistant materials, although other genes did not show a similar pattern (Figure 7B).

Finally, 12 DEGs were selected to perform qRT-PCR to verify the accuracy of the RNA-Seq data (Figure 8). Statistical analysis of the differences in phenotypic data between the extreme materials used for qRT-PCR indicated that source-/sink-related yield traits, such as seed weight per silique index (SPSI), seed yield per plant (SY), and harvest index (HI) that refers to the ratio of economic yield to biological yield and reflects the transportation ability of the photosynthate from source to sink organs, were significantly different between low-yield-related material (L1) and high-yield-related material (L2).

## 3. Discussion

### 3.1. Genome-Wide Identification and Phylogenetic and Syntenic Analysis of TPSs

*B. napus* (AACC, 2n = 38), an important oilseed crop worldwide, is a heterogeneous hybrid derived from two diploid species (*B. oleracea*, n = 9, and *B. rapa*, n = 10) [32]. The relationships among the three species are described by the “U’s triangle” model [33]. Multiple comparative genomic analyses between *B. rapa* and the model plant *A. thaliana* revealed that a whole genome triplication (WGT) event occurred millions of years ago [34,35]. The origin and evolution of *Brassica napus* has become an important focus of study. Numerous studies have focused on genome-wide identification and evolutionary analysis of various gene families in *B. napus*, such as monosaccharide transporter genes (*MST*) [33], Gretchen Hagen 3 (*GH3*) [36], cytokinin oxidase/dehydrogenase (*CKX*) [37], fatty acid desaturase (*FAD*) [38], and glutathione transferase (*GST*) genes [39]. However, to date, no studies have been performed on the genome-wide identification and expression patterns of *TPS*s in this crop. In the current study, we identified 14 *BoTPS*s, 17 *BrTPS*s, and 35 *BnTPS*s in *Brassica* based on the 11 *AtTPS* and 1 *AtTRE* sequences in Arabidopsis (Figure 1, Table 2). These numbers differ from those expected from the WGT event, i.e., 36 *BoTPS*s, 36 *BrTPS*s, and 72 *BnTPS*s, indicating that not all gene families fit the numbers expected from the WGT event and that genome shrinkage or redundancy occurred in some *TPS* gene families during the long history of evolution [33,40].

A phylogenetic tree of *TPS*s among *A. thaliana* and the three Brassica species was constructed. All *TPS*s were separated into three subfamilies: subfamily A (*TPS1* to *TPS4*), B (*TPS5* to *TPS11*), and C (*TRE*) (Figure 1). Syntenic analysis was performed to clarify the evolutionary relationships of the *TPS*s among the four species, and 78 homologous pairs were identified (Figure 2A). Collinearity analysis among the genes from the A (Figure 2B) and C (Figure 2C) sub-genomes revealed 46 and 41 homologous pairs, respectively. In addition, most *BnTPS*s (*BnTPS2*, *BnTPS3*, *BnTPS4*, *BnTPS6*, *BnTPS8*, *BnTPS9*, and *BnTPS11*) were present in two copies, except for *BnTRE* (five copies), and the other genes were present in four copies. In short, genome shrinkage occurred during the evolution of *TPS* genes. Similar results were obtained in previous studies, suggesting that genome shrinkage and redundancy might have occurred after the WGT event [33,40]. In addition, several recent studies suggested that gene loss or the generation of multiple copies occurred during evolution [33,41,42].

Based on phylogenetic analysis and the protein profiles of the predicted proteins of the 35 *BnTPS*s, it was concluded that genes from the same class share similar features. For instance, genes from subfamily A contain more exons than those from the other two subfamilies, especially subfamily B (Figure 4). *Bn*TPSs in subfamilies A and B contain more predicted conserved motifs than those in subfamily C, and some distinctive motifs were identified in subfamilies A and B, which could contribute to the functional differentiation between these subfamilies (Figure 5). Some predicted proteins possess significantly different profiles even though they belong to the same class. Twenty-two identical motifs were predicted in both BnTPS5-1 and BnTPS5-2, but only three were predicted in both BnTPS5-3 and BnTPS5-4, even though all four genes are members of the *BnTPS5* subgroup (Figure 5). The same situation was observed in *BnTPS7*, with *BnTPS7-1*, *BnTPS7-2*, *BnTPS7-3*, and *BnTPS7-4* exhibiting completely different gene structures and conserved motifs (Figure 5). These obvious differences among genes in the same class were likely due to genome loss, resulting in functional differentiation.

Genome-wide identification and phylogenetic and syntenic analyses suggested that gene elimination and loss occurred during the genomic evolution of *BnTPS*s after the WGT event. The profiles of many homologous genes also changed during the process of genomic evolution, such as gene structures and conserved motifs, which might further alter their expression patterns and functions [43].

### 3.2. Expression Pattern Analysis and Functional Prediction of BnTPSs

*TPSs* play crucial roles in both plant development and protection from abiotic stresses, such as drought [27,44,45]. Hence, to select key genes for further functional study among the 35 *BnTPS*s, expression data from *B. napus* cultivar ZS11 on BrassicaEDB (https://brassica.biodb.org/index, accessed on 30 November 2022) [46] was acquired to analyze the spatiotemporal expression patterns of the 35 *BnTPS*s because the expression specificity of a gene is likely associated with its function (Figure 6). Two sets of transcriptome data of the 35 *BnTPS*s were also analyzed and visualized: one data set from the yield-related materials examined in our laboratory (Figure 7A) and the other from drought-related materials downloaded from BrassicaEDB under NCBI SRA project ID PRJNA270960 [47] (Figure 7B). Additionally, qRT-PCR of 12 randomly selected DEGs was performed to verify the accuracy of our RNA-Seq data. The results of qRT-PCR were basically consistent with the RNA-Seq data (Figure 8).

Most *BnTPS*s showed low expression levels, but some showed significant temporal and spatial expression specificity, and most but not all *BnTPS*s shared similar expression profiles based on their phylogenetic classifications (Figure 6 and Figure 7). For example, *BnTPS*s in subfamily B had obviously complex expression patterns compared with genes from subfamily A. Similarly, genes in subfamilies A and C were barely expressed in the drought-related materials, whereas the expression of many genes in subfamily B, such as *BnTPS6-2*, *BnTPS8-1*/*8-2*, *BnTPS9-1*/*9-2*, *BnTPS10-3*, and *BnTPS11-1*/*11-2*, increased sharply in drought-resistant materials after drought stress (Figure 7B).

All 35 *BnTPS*s could be divided into two categories based on their expression characteristics. However, some genes exhibited different expression profiles in the yield-related materials. *BnTPS5-1*, *BnTPS5-2*, *BnTPS1-2*, and *BnTPS1-3* were expressed at higher levels in sink (seeds) than in source (pericarps) organs, whereas the expression levels of *BnTPS8-2*, *BnTPS9-1*, and *BnTPS9-2* were lower in sink than in source organs. T6P acts as a signal that regulates carbon allocation and serves as a target to improve crop yields [48], and TPS and TPP genes are associated with source-/sink-related yield traits [49]. As described above, genes like *BnTPS1*, *BnTPS5*, and *BnTPS9* exhibited different expression levels between source and sink organs, and the materials we used to perform RNA-Seq, L1 and L2, showed significantly different seed yield, harvest index, SPSI, and sucrose content between seeds and pericarps (Figure 9). Sucrose is the main product of photosynthesis, and differences in sucrose content between seeds and pericarps indicate the differences in the transport ability of the photosynthate from source (pericarp) to sink (seed) organs, which may lead to differences in yield. These results suggest that six *BnTPS*s (*BnTPS1-2*, *BnTPS1-3*, *BnTPS5-1*, *BnTPS5-2*, *BnTPS9-1*, *BnTPS9-2*) are highly associated with source-/ sink-related yield traits in *B. napus* (Figure 10).

However, numerous studies have revealed that *TPS*s play important roles in plant desiccation resistance as well as responses to other abiotic stimuli [1,44,50]. In the current study, some of the 35 *BnTPS*s were barely expressed in the drought-resistant/-sensitive material, but the expression of several genes, such as *BnTPS6-2*, *BnTPS8-1*, *BnTPS8-2*, *BnTPS9-1*, *BnTPS9-2*, *BnTPS11-1*, and *BnTPS11-2*. Increased sharply after drought treatment. More important, the expression of these genes increased more in drought-resistant materials than in drought-sensitive materials, suggesting that the seven *BnTPS*s listed above might be associated with desiccation resistance in *B. napus* (Figure 10).

In summary, the expression patterns of *BnTPS*s vary according to their subfamily classifications, but some *BnTPS*s show different expression patterns even though they belong to the same subfamily. These results suggest that homologous genes do not necessarily perform similar functions [33]. By combining phylogenetic analysis, protein profile prediction, and expression pattern analysis of the 35 *BnTPS*s, it was determined that *TPS* genes in *B. napus* suffered serious gene loss during the process of genome-wide evolution, and the changes in gene structure and protein characteristics resulted in the differential expression. Genes encoding proteins with more motifs might have higher expression levels, which may ultimately cause functional differentiation.

## 4. Materials and Methods

### 4.1. Identification of TPSs in B. napus, B. rapa, and B. oleracea

The 12 deduced protein sequences of *AtTPS*s were downloaded from the Arabidopsis Information Resource database (https://www.arabidopsis.org, accessed on 13 October 2020) [51,52]. The *BnTPS*s, *BoTPS*s, and *BrTPS*s were identified using BLASTp analysis based on *At*TPS protein sequences as well as the *At*TPS Pfam numbers [53,54] and HMMsearch program (https://www.ebi.ac.uk/Tools/hmmer/, accessed on 13 October 2020) [55].

### 4.2. Multiple Sequence Alignment and Phylogenetic and Syntenic Analysis

To clarify the evolutionary relationships of the *TPS*s among the four major cruciferous plants, multiple sequence alignment and phylogenetic analysis were performed. Molecular Evolutionary Genetics Analysis (MEGA) 7.0 software was used to perform multiple sequence alignment with default parameters, and a phylogenetic tree with 1000 bootstrap replicates was constructed using the neighbor-joining (NJ) method [56]. The FigTree tool was used to visualize the phylogenetic tree. Syntenic analysis was performed using Tbtools software [57], which revealed the synteny relationships of *TPS* genes from *B. napus* and the three other species.

### 4.3. Chromosomal Locations, Gene Structures, and Conserved Motifs of BnTPSs

To further explore the characteristics of *BnTPS*s, the chromosomal locations, gene structures, and conserved motifs were analyzed. Detailed information about the locations of the *BnTPS*s was obtained from the *Brassica napus* Genome Browser (https://www.genoscope.cns.fr/brassicanapus/cgi-bin/gbrowse/colza/, accessed on 27 March 2022) [58], and the figure showing chromosomal distribution was created using MapChart2.2 [59]. The gene structures and conserved motifs of *BnTPS*s were analyzed using Gene Structure Display Server (GSDS) [60] and Multiple Em for Motif Elicitation (MEME) (http://meme-suite.org/tools/meme, accessed on 28 November 2020) [61,62], respectively: the maximum number of predicted motifs was set to 25, and the default settings were used for the other parameters.

The isoelectric point (pI) and molecular weight (Mw) of each *Bn*TPS was predicted according to the ExPASy proteomics server database (https://www.expasy.org/tools/, accessed on 3 February 2021) [63]. The subcellular localizations of the *Bn*TPS proteins were predicted online with MultiLoc2 (https://abi-services.informatik.uni-tuebingen.de/multiloc2/webloc.cgi, accessed on 3 February 2021) [64].

### 4.4. Plant Materials and Phenotyping

Numerous reports indicate that *TPS* genes are highly associated with source-/sink-related yield and drought response. Therefore, we collected seeds and pericarps of L1 and L2, which represent a pair of materials with extremely high and low SPSI (seeds weight per silique index), respectively, reflecting the carbohydrate distribution between source and sink organs, which influences source-/sink-related yield traits.

The seeds of L1 and L2 were obtained from Chongqing Rapeseed Technology Research Center, China, and germinated in seed plots under natural environments in September. All the plants were transplanted to the experimental fields in about a month with 10 plants per row. All materials were cultivated under the same conditions. We recorded the flowering date after the materials bloomed and collected seeds and silique pericarps at 15 and 35 DAF. The samples were immediately frozen in liquid nitrogen and stored at −80 °C until use. The phenotypes of the plants analyzed in this study are shown in Figure 9, and the yield-related phenotypic data of L1 and L2 are shown in Table 3.

### 4.5. RNA-Seq Analysis and qRT-PCR Verification

To identify the major candidate genes among the 35 *BnTPSs* that might be related to the seed yield of *B. napus*, a set of transcriptome data generated in our laboratory was analyzed, which was obtained from source (silique pericarps from the main branch) and sink tissues (seeds from the main branch) from two materials (L1 and L2) with extremely high and low yields, respectively. The expression data of drought-resistant/-sensitive materials (PRJNA270960) and cultivar ZS11 were downloaded from BrassicaEDB (https://brassica.biodb.org/index, accessed on 30 November 2022) [46]. Subsequently, the heat map was visualized using TBtools [57].

To verify the RNA-Seq data, RNA was extracted from the materials mentioned above using an RNeasy Extraction Kit (Invitrogen, Carlsbad, CA, USA) to produce cDNA with a Reverse Transcription Kit (TaKaRa Biotechnology, Dalian, China). qRT-PCR was performed to validate the RNA-Seq data as described by Qu [65]. Premier 5.0 was used to design specific primers for qRT-PCR [66], and the primers were further confirmed with online tools (BRAD, http://brassicadb.cn, accessed on 25 January 2021) [67]. *UBC21* was chosen as the endogenous reference gene [68]. All primers used for qRT-PCR are shown in Appendix A. The relative expression levels of the candidate genes were calculated using the 2^−ΔΔCt^ method [69]. The results were visualized with GraphPad Prism software [70].

## 5. Conclusions

Trehalose is important for plant development, and it plays an essential role in response to abiotic stress. TPSs participate in the trehalose metabolism and form a large family with multiple copies showing extensive functional diversification. In this study, we systematically analyzed the *TPS* gene family among four major Cruciferae species using genome-wide identification, phylogenetic and syntenic analyses, protein profile analysis, and RNA-Seq analysis of yield- and drought-related materials. Genome-wide identification and phylogenetic analysis among the four species indicated that only gene elimination occurred during the evolution of *BnTPS*s. By combining protein profile prediction and RNA-Seq analysis of materials with extremely high and low yields, we determined that the 35 identified *BnTPS*s can be divided into two major functional classes: one responding to drought stress (such as *BnTPS6*, *BnTPS8*, *BnTPS9*, and *BnTPS11*), and the other functioning in source-/sink-related yield traits (such as *BnTPS1*, *BnTPS5*, and *BnTPS9*). The *BnTPS*s that were identified in this study underwent gene elimination leading to changes in their profiles during genome evolution, which may have influenced their expression patterns and caused functional divergence. Our study provided basic information about *TPS*s in *Brassica napus* and identified several candidate genes related to both yield and drought resistance.

## Figures and Tables

**Figure 1 plants-12-00981-f001:**
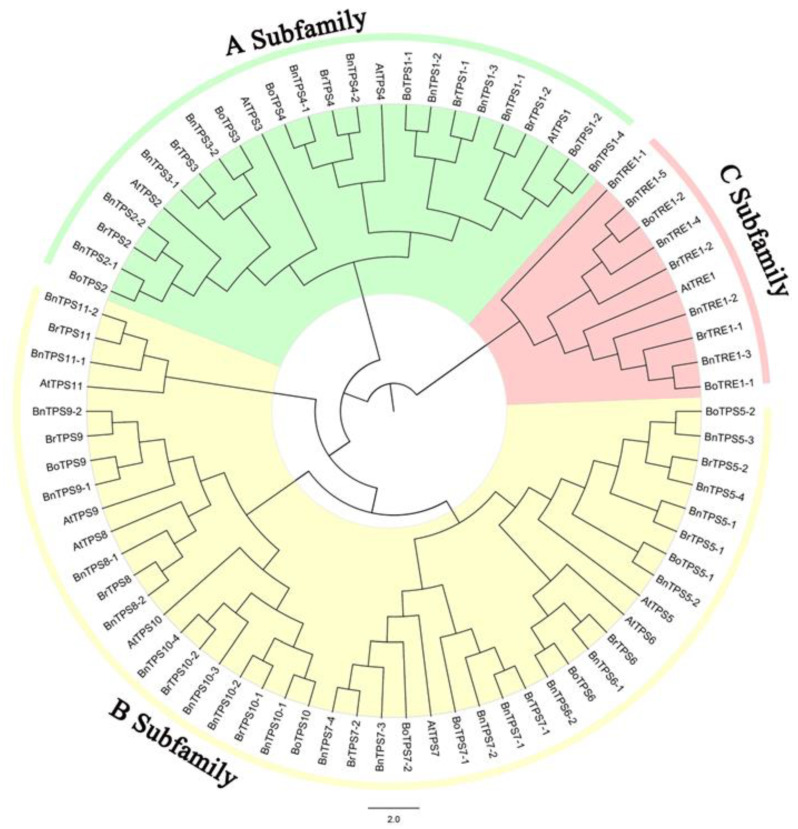
Neighbor-Joining (NJ) tree of the Trehalose-6-phosphate synthase (TPS) protein sequences from *Brassica napus*, *Brassica oleracea*, *Brassica rapa*, and *Arabidopsis thaliana*. All *TPS* genes were divided into 3 subfamilies, named A to C.

**Figure 2 plants-12-00981-f002:**
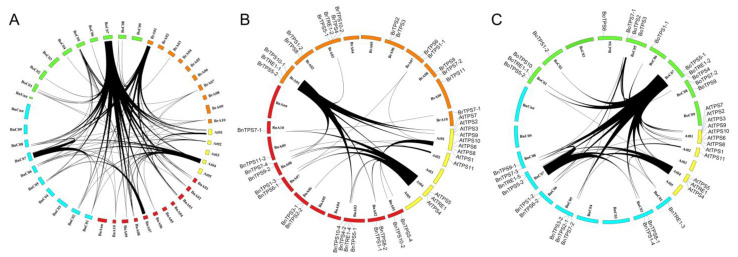
Syntenic relationships of *TPS* genes, as indicated by connecting lines. (**A**) Syntenic relationships of the total 78 *TPS*s among the 4 species. (**B**) Syntenic relationships of *TPS*s among the Arabidopsis, *B. rapa*, and subgenome A from *B. napus*. (**C**) Syntenic relationships of *TPS*s among the Arabidopsis, *B. oleracea*, and subgenome C from *B. napus*.

**Figure 3 plants-12-00981-f003:**
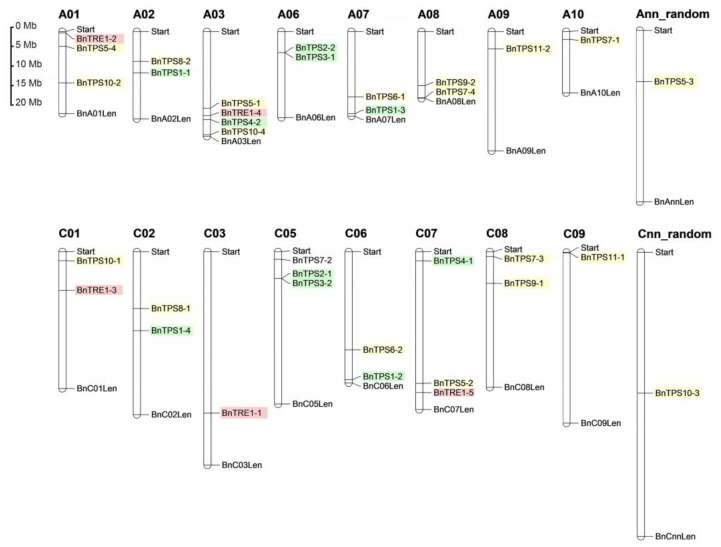
The chromosomal location of the 35 *BnTPS*s. The background color corresponding to the phylogenetic tree indicated different subfamilies.

**Figure 4 plants-12-00981-f004:**
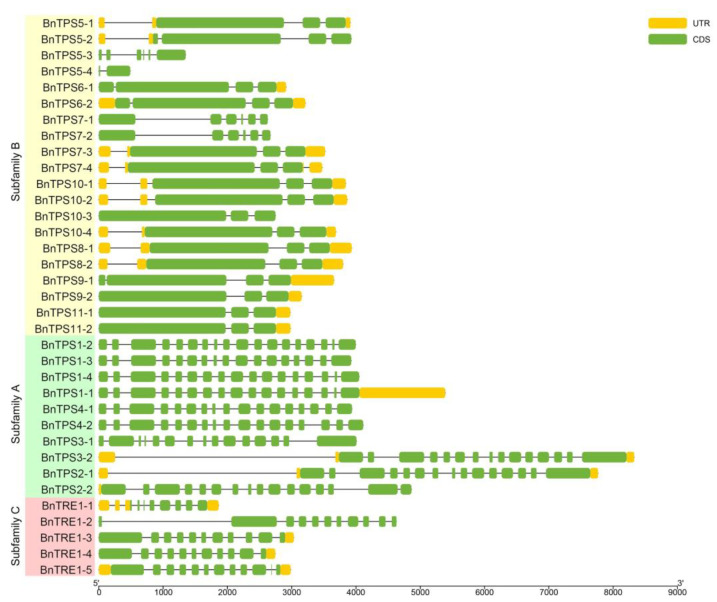
The gene structures of the 35 *BnTPS*s based on their phylogenetic relationships.

**Figure 5 plants-12-00981-f005:**
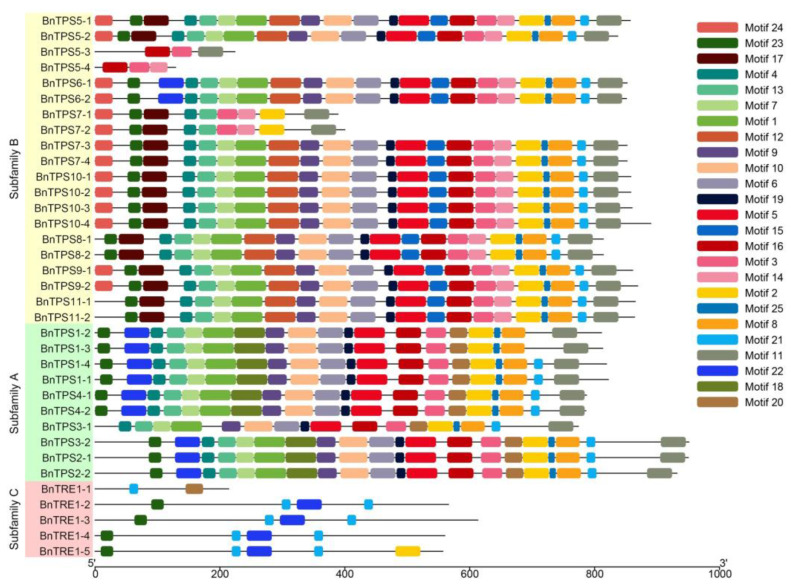
The conserved motifs of the 35 *BnTPS*s based on their phylogenetic relationships.

**Figure 6 plants-12-00981-f006:**
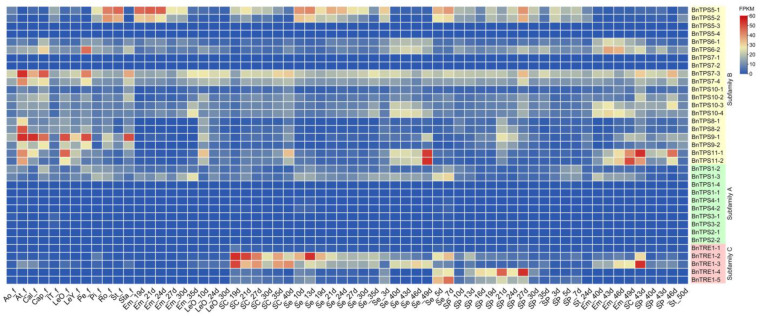
Heat map of the expression profiles of the 35 *BnTPS*s in various tissues of ZS11. Ao: anthocaulus; At: anther; Cal: calyx; Cap: capillament; IT: top of inflorescence; LeO: mature leaves; LeY: young leaves; Pe: petal; Pi: pistil; Ro: roots; St: stems; Sta: stamens; Em: embryo; SC: seed coat; Se: seeds; SP: silique pericarps; f: full-bloom stage. The number means the corresponding days after flowering.

**Figure 7 plants-12-00981-f007:**
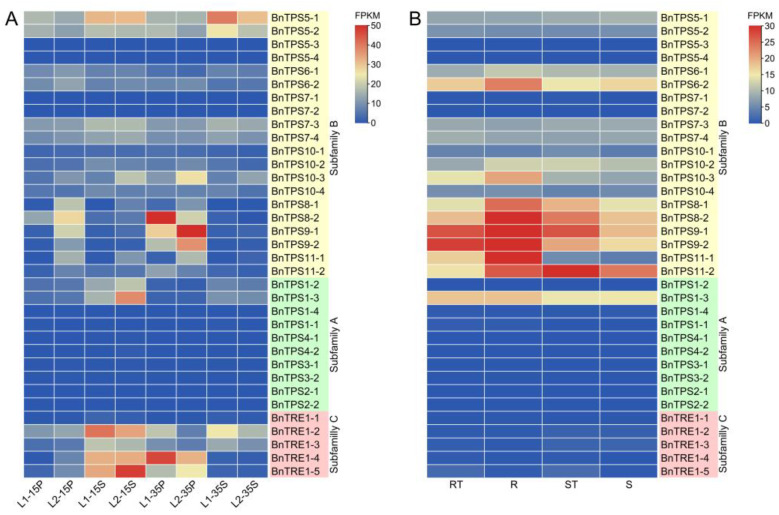
Heat map of the expression profiles of the 35 *BnTPS*s among different materials in various tissues. (**A**) The expression in yield-related materials; (**B**) The expression in drought-related materials. L1: Low yield-related material; L2: High yield-related material; RT: Drought-resistant material; R: RT treated with PEG-6000; ST: Drought-sensitive material; S: ST treated with PEG-6000. S: seeds; P: pericarps. The number of 15/35 means 15/35 days after flowering.

**Figure 8 plants-12-00981-f008:**
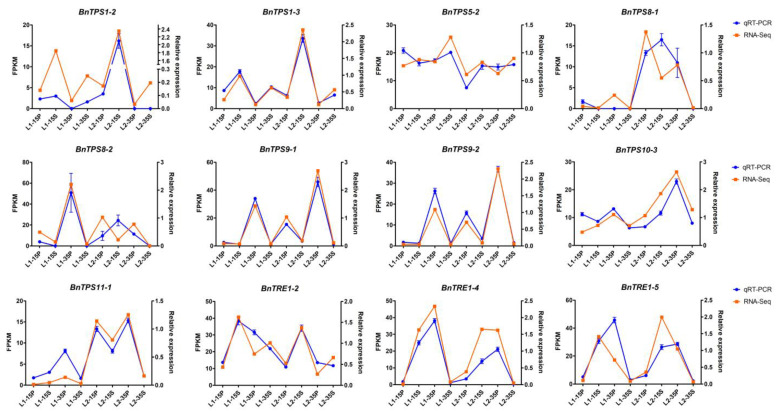
qRT-PCR verification of the 12 DEGs in diverse tissues and growth periods between materials with extremely high (L2) and low (L1) SPSI; P: silique pericarps; S: seed.

**Figure 9 plants-12-00981-f009:**
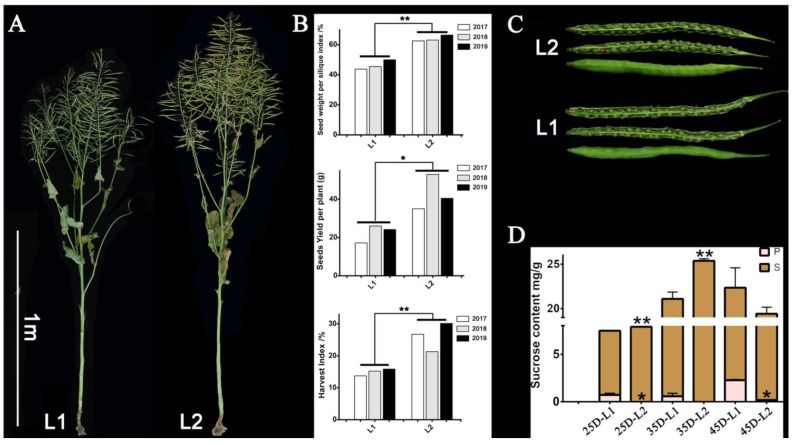
The phenotype of the plant materials associated with source-/sink-related yield traits and drought response. (**A**) The plant phenotype of materials with extremely high (L2) and low (L1) yield traits. (**B**) The statistical analysis of source-/sink-related yield traits between materials of L1 and L2. (**C**) The silique phenotype of materials with extremely high (L2) and low (L1) yield traits. (**D**) Dynamic content of sucrose in source-/sink-tissues of extreme materials during different developmental stages. L1: Low yield-related material; L2: High yield-related material; The number of 15, 25, 35, and 45 mean corresponding days after flowering; S: seeds; P: pericarps; **: significant difference at *p* < 0.01; *: significant difference at *p* < 0.05.

**Figure 10 plants-12-00981-f010:**
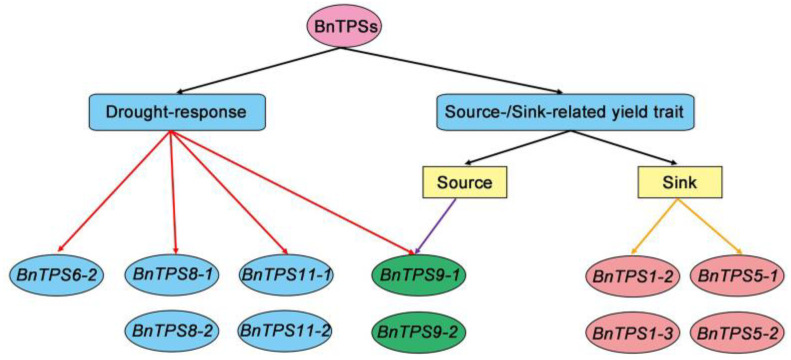
Functional classification of *BnTPS*s during evolution according to the expression patterns.

**Table 1 plants-12-00981-t001:** Homologous genes in the A and C sub-genomes of *B. napus*, *B. rapa*, *B. oleracea*, and Arabidopsis.

Gene ID (At)	Gene Name (At)	Gene ID (Bra)	Gene ID (Bol)	Gene ID (BnaA)	Gene ID (BnaC)
*AT1G78580*	*AtTPS1*	*Bra035049*	*Bol027474*	*BnaA02g18920D*	*BnaC06g39000D*
		*Bra008366*	*Bol018953*	*BnaA07g34230D*	*BnaC02g25020D*
*AT1G16980*	*AtTPS2*	*Bra026011*	*——*	*BnaA06g11450D*	*BnaC05g13090D*
*AT1G17000*	*AtTPS3*	*Bra026010*	*Bol038272*	*BnaA06g11460D*	*BnaC05g13110D*
*AT4G27550*	*AtTPS4*	*Bra019043*	*Bol042340*	*BnaA03g48650D*	*BnaC07g50330D*
*AT4G17770*	*AtTPS5*	*Bra012642*	*Bol037106*	*BnaA03g43320D*	*BnaC07g34770D*
		*Bra040180*	*Bol019698*	*BnaAnng13570D*	*——*
		*——*	*——*	*BnaA01g08660D*	*——*
*AT1G68020*	*AtTPS6*	*Bra004054*	*Bol027823*	*BnaA07g24830D*	*BnaC06g26160D*
*AT1G06410*	*AtTPS7*	*Bra015497*	*Bol041057*	*BnaA10g04170D*	*BnaC05g04390D*
		*Bra030651*	*Bol023345*	*BnaA08g28610D*	*BnaC08g01800D*
*AT1G70290*	*AtTPS8*	*Bra007906*	*——*	*BnaA02g14790D*	*BnaC02g19750D*
*AT1G23870*	*AtTPS9*	*Bra016328*	*Bol008681*	*BnaA08g20280D*	*BnaC08g06450D*
*AT1G60140*	*AtTPS10*	*Bra031526*	*Bol036586*	*BnaA01g22200D*	*BnaC01g43070D*
		*Bra017888*	*——*	*BnaA03g54540D*	*BnaCnng41250D*
*AT2G18700*	*AtTPS11*	*Bra038548*	*Bol038270*	*BnaA09g09720D*	*BnaC09g51060D*
*AT4G24040*	*AtTRE1*	*Bra013756*	*Bol009586*	*BnaA01g35070D*	*BnaC03g56490D*
		*Bra019249*	*Bol042146*	*BnaA03g46430D*	*BnaC01g15870D*
		*——*	*——*	*——*	*BnaC07g38690D*

**Table 2 plants-12-00981-t002:** The number of homologous genes in the A and C sub-genomes of *B. napus*, *B. rapa*, *B. oleracea*, and Arabidopsis.

Gene Name	*A. thaliana*	*B. rapa*	*B. oleracea*	*B. napus* (A)	*B. napus* (C)
Two-Copy in *B. napus*					
*TPS2*	1	1	0	1	1
*TPS3*	1	1	1	1	1
*TPS4*	1	1	1	1	1
*TPS6*	1	1	1	1	1
*TPS8*	1	1	0	1	1
*TPS9*	1	1	1	1	1
*TPS11*	1	1	1	1	1
Four-Copy in *B. napus*					
*TPS1*	1	2	2	2	2
*TPS5*	1	2	2	3	1
*TPS7*	1	2	2	2	2
*TPS10*	1	2	1	2	2
Five-Copy in *B. napus*					
*TRE1*	1	2	2	2	3
TOTAL	12	17	14	18	17

**Table 3 plants-12-00981-t003:** Yield-related phenotypic data of plant materials during 3 years of field tests.

Trait	Material	2017	2018	2019	Mean Value	SEM	*p*-Value
Seeds weight per silique index/% (SPSI)	L1	43.73	45.49	49.95	46.39	2.62	0.0013
	L2	62.57	63.10	66.38	64.02	1.68	
Seeds Yield per plant/g (SY)	L1	17.19	26.00	24.20	22.46	3.80	0.0267
	L2	35.04	53.00	40.50	42.85	7.52	
Harvest index/% (HI)	L1	13.75	15.24	15.89	14.96	0.90	0.0135
	L2	26.73	21.36	30.15	26.08	3.62	

## Data Availability

All data is contained within the article and Appendix A.

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
