# Peer review of "Identification of Trehalose-6-Phosphate Synthase (TPS) Genes Associated with Both Source-/Sink-Related Yield Traits and Drought Response in Rapeseed (Brassica napus L.)"

_plants, 2023, doi:10.3390/plants12050981_

Round 1

Reviewer 1 Report

The subject of study is interesting considering the attention Trehalose signaling has gotten in the last few years. It has also emerged as an important candidate that controls traits that are economically relevant. The manuscript is well-written and the data is nicely presented. I have the following minor recommendations that can be addressed by the authors and resubmitted for acceptance-

Line 27- consider changing the wording to “exhibited variable expression”

Line 52- references for Populus [14, 15] refers to [15,16] instead

Line 53- That binomial nomenclature Solanum tuberosum is for Potato! It says in the title of the reference cited!!

Line 109- Figure 2 Please consider increasing the font size in the figures. This gets pixelated when zoomed in and is hard to follow.

Line 120- Table-2 Scientific name should always be in italics. B. napus

Line 127- Figure 3 Please consider increasing the font size in the figures. This gets pixelated when zoomed in and is hard to follow.

Line 163- "......and was BnTPS7-3" - not clear what authors want to say here.

Line167- Please consider changing the wording of “strong functional differentiation during…..” to “strong functional divergence during ......“

Line 180- Please refer to the figure/data source at the end of the sentence.

Line 184- Please clarify the phrase "yield-related materials".

Line 216- “Harvest Index”- Please consider adding a phrase to explain this term for general readers who may not be acquainted with this system.

Line 219- Figure 8. How do the authors explain the difference in peaks for values for BnTPS8-1 and BnTPS8-1 from qPCR and RNAseq here?

Line 240- Please provide references here at the end of the sentence (if such proposition is a novel hypothesis by the authors first reported based on this study, or a similar connection has been found to be true in other species/other studies in the above species based on other gene families). It's discussed later, but the reference should go here too.

Line 257- Please be consistent in notation throughout the manuscript. Bn

Line 298- …..allocation and serves as

Lines 302—303 “and sucrose content between seeds and pericarp”….Please add a line as to why this feature is vital in this TPS gene context.

Line 330- Scientific name should always be in italics.  B. napus, B. rapa, and B. oleracea

Line 333—334- Please be consistent in notation throughout the manuscript. AtTPS 

Line 348- Scientific name should always be in italics. Brassica napus 

Line 357- Please be consistent in notation throughout the manuscript. Bn

Line 366- Please mention the growth conditions for the plants that were used for collecting tissues (Figure 10), and for other parameters if different sets of plants were used for field trial (Table 3). 

Start with seed wash/treatment to transplanting (if done for field conditions), the time of growth (month/season), average temperature (day/night), light condition, watering/fertilizer schedule, relative humidity, light intensity (if grown in greenhouse/growth chambers)….. This aids in data transparency.

References- Please maintain uniform formatting for the references including 33, 35, 37, 38, 42, 45, 47, 51, 53, 54, and 62 (please refer to the attached *.pdf file for reference).

Author Response

We are very grateful to you for reviewing our manuscript carefully. Please see the attachment.

Reviewer 2 Report

Manuscript titled "Identification and Expression Profiling of Trehalose-6-phosphate Synthase (TPS) Genes Associated with Both Source-/Sink-Related Yield Traits and Drought Response in Rapeseed (Brassica napus L.)" contained some new information for the researchers. Author claimed that no genome-wide identification or functional prediction of TPSs has been performed in B. napus.

Many sentences in the manuscript need revision e.g. Line 76, 84, 104, 106, 108, 115, 131, 152, 176, 241, 255, and many other. Change these sentences in passive voice.

 Too many references from the website that should be replaced with the authentic source.

Acknowledgements???

Introduction section has well written with the sufficient citations of references.  

Results: Font size of some figures are too small and not coinvent to read.

Author should first present the result of the present study and then discuss it and take a support from literature. Line 88, 

Presentation of the results must be improved and briefed. 

Discussion needs improvement. Line 268 Conclusion?

Materials and Methods

Plant material detail is missing.

Too much figures and some can be removed.

Conclusion section must be improved, and logical conclusion should be given.

Author Response

(The authors gave the same response as above.)
